# Contribution of Rare and Common *APOE* Variants to Familial Hypercholesterolemia in Spanish Cohort

Lorena M. Vega-Prado [1], Daniel Vázquez-Coto [2], Francisco Villazón [3], Lorena Suárez-Gutiérrez [3], Ceferino Martínez-Faedo [3], Edelmiro Menéndez-Torre [2,3], María Riestra [4], Silvia González-Martínez [4], Gala Gutiérrez-Buey [4], Claudia García-Lago [2], Juan Gómez [1,2,5,6], Victoria Alvarez [1,2], Helena Gil [1,2], Rebeca Lorca [2,5,6,7,8] and Eliecer Coto [1,2,5,6,8,*]

[1] Servicio de Genética, Hospital Universitario Central Asturias (HUCA), 33011 Oviedo, Spain; lorenamaria.vega@sespa.es (L.M.V.-P.); juan.gomezde@sespa.es (J.G.); victoria.alvarez@sespa.es (V.A.); hgilpena@gmail.com (H.G.)

[2] Instituto de Investigación Sanitaria del Principado de Asturias (ISPA), 33011 Oviedo, Spain; teledaniel.22@gmail.com (D.V.-C.); edelmiro.menendez@sespa.es (E.M.-T.); lorcarebeca@gmail.com (R.L.)

[3] Servicio de Endocrinología y Nutrición, Hospital Universitario Central Asturias (HUCA), 33011 Oviedo, Spain; fvillazon@yahoo.es (F.V.); loresuarezgu@gmail.com (L.S.-G.); ceferino.mfaedo@sespa.es (C.M.-F.)

[4] Servicio de Endocrinología y Nutrición, Hospital Universitario de Cabueñes, 33394 Gijón, Spain; mriestra.fernandez@gmail.com (M.R.); silvia311289@gmail.com (S.G.-M.); galagutierrezbuey@gmail.com (G.G.-B.)

[5] Unidad de Cardiopatías Familiares, Hospital Universitario Central Asturias (HUCA), 33011 Oviedo, Spain

[6] Redes de Investigación Cooperativa Orientadas a Resultados en Salud (RICORs), 28014 Madrid, Spain

[7] Área del Corazón, Hospital Universitario Central Asturias (HUCA), 33011 Oviedo, Spain

[8] Departamento Medicina, Universidad de Oviedo, 33011 Oviedo, Spain

* Correspondence: eliecer.coto@sespa.es

**Abstract:** Our aim was to determine whether rare *APOE* pathogenic variants (PV) and the common e2/e3/e4 polymorphism were associated with the risk of familial hypercholesterolemia (FH). A total of 431 patients who met the inclusion criteria for FH were next-generation sequenced for the main candidate genes (*LDLR, APOB, PCSK9, APOE, LDLRAP1*). A total of 139 patients (32%) had a pathogenic variant, including 3 with *APOE* p.Leu167del. Among the PV-negatives (n = 292), one was homozygous for *APOE*-e2 and showed a combined phenotype of high low-density lipoprotein cholesterol (LDL-C) and triglycerides (TGs). A total of 165 population controls were also genotyped for the *APOE* polymorphism. PV-negative patients showed a significantly higher frequency of *APOE*-e3e4/e4e4 compared to PV-positives (*p* = 0.006) and to population controls (*p* = 0.0002, OR = 2.63, 95% CI = 1.57–4.40). *APOE*-e4e4 patients had significantly higher mean LDL-C compared to the other genotypes (*p* = 0.047). In conclusion, *APOE* pathogenic variants were a rare cause of FH in our population, and the *APOE*-e4 allele was a significant risk factor for being diagnosed with familial hypercholesterolemia in the absence of a pathogenic variant involved in FH. In particular, the *APOE*-e4e4 genotype was associated with higher LDL-C levels compared to the other genotypes.

**Keywords:** familial hypercholesterolemia; apolipoprotein E; polymorphisms; genetic association

## 1. Introduction

Familial hypercholesterolemia (FH) is a genetic disorder characterised by life-long elevated levels of cholesterol and low-density lipoprotein cholesterol (LDL-C). FH has an

estimated prevalence of about 1 in 200 people and is associated with an increased risk of premature coronary and cerebral ischemic diseases. These patients require treatment with lipid-lowering drugs to reduce the risk for these events and normalise life expectancy.

Many FH patients harbour a highly penetrant pathogenic variant (PV) in one of the recognised FH genes, inherited as dominant (*LDLR*, *APOB*, *APOE*, *PCSK9*) or biallelic/recessive (*LDLRAP1*) [1–5]. In 30–50% of these FH patients, a PV is not identified, and they canthus carry low-penetrant variants associated with a moderate increase in the LDL-C levels in the general population [6–8]. The main candidate genes for polygenic FH are also the same that showed highly penetrant PVs, such as *LDLR*, *APOB*, *APOE*, and *PCSK9*. Several common polymorphisms (SNPs) at these genes have been related to the risk of presenting LDL-C above the normal range in the general population, being thus more frequent in individuals with hypercholesterolemia. Based on a number of candidate SNPs, several polygenic risk scores have been proposed as predictors of LDL-C levels and the risk of atherosclerotic disease [9–16]. Among these, the common *APOE*-e2/3/4 has been widely associated with total cholesterol and LDL-C levels in the general population, with *APOE*-e4 carriers showing higher levels compared to non-carriers. This *APOE* variant could thus contribute to the risk of presenting analytical and clinical features of FH in the absence of recognised PVs.

The main aim of our study was to determine the association between FH and common *LDLR*, *APOB*, *PCSK9*, and *APOE* polymorphisms.

## 2. Patients, Controls, and Methods

### 2.1. Study Population

We studied a total of 431 non-related patients (index cases, aged > 18 years) with probable FH according to the Dutch Lipid Clinical Network (DLCN) criteria (Table 1). The basal lipid values were those corresponding to the pre-treatment with lipid-lowering drugs, as it was collected in the clinical history of the patients. The controls were a total of 165 normo-lipaemic individuals (LDL-C < 150 mg/dL) aged 18–60 years and from the same patient's region (Asturias, Northern Spain, total population 1 million). This study was approved by the Ethical Committee for medical Research, and all the participants signed an informed consent form.

**Table 1.** Main characteristics of the study patients.

| | FH Patients N = 431 | Controls N = 165 |
|---|---|---|
| **Men** | 227 (53%) | 78 (47%) |
| **Women** | 204 (47%) | 87 (53%) |
| **Age at diagnostic range years** | 18–45 | 18–60 |
| **Total cholesterol mG/dL mean (range)** | 345 (275–926) | 205 (78–310) |
| **LDL-cholesterol mean (range)** | 248 (180–750) | 132 (61–150) |
| **Triglycerides** | 194 (15–1132) | 97 (28–320) |
| **DLCN score** | | |
| **3–5 possible** | 25% | - |
| **6–8 probable** | 40% | - |
| **>8 definite** | 35% | - |

DLCN = Dutch clinical lipid network.

### 2.2. Genetic Testing for Pathogenic Variants

We obtained the DNA from blood leukocytes from patients and controls. All the patients were evaluated for the presence of pathogenic variants related to FH in the *LDLR*, *APOB*, *PCSK9*, *LDLRAP1*, and *APOE* genes. We used next-generation sequencing (NGS) with semiconductor chips (IonTorrent technology, Fisher Scientific, Waltham, MA, USA) to sequence the coding exons, plus at least 10 intronic flanking nucleotides of the 5 genes. An Ion Reporter (Thermo Fisher Scientific, Waltham, MA, USA) and HD Genome One (DREAMgenics S.L., www.dreamgenics.com (28 December 2024), Oviedo, Spain) software were used for variant annotation and classification (pathogenicity) according to the American College of Medical Genetics and Genomics (ACMG-AMP) Standards and Guidelines [17]. The detailed technical procedure has been previously reported [18].

We did not determine major rearrangements (mainly large deletions) of the LDLR gene, which have been described in approximately 10% of patients with severe familial hypercholesterolemia.

### 2.3. APOE Genotyping

All the patients and controls were studied for the *APOE*-e2/e3/e4 variants, defined by SNPs rs429358 and rs7412, that result in two amino acid changes at protein positions 130 (Cys in e3 and e2, and Arg in e4) and 176 (Arg in e3, Cys in e2, and Arg in e4). The two SNPS were determined by real-time PCR with Taqman probes (assays ID C_3084793_20 and C_904973_10) in ABI7500 equipment (Fisher Scientific, Waltham, MA, USA). The quality of this method was demonstrated by the concordance with the genotypes determined by NGS in the patients.

### 2.4. Statistical Analysis

All the patient and control values were collected in an excel file: among others, this included sex, age, LDL-C pre-treatment level, type of pathogenic variant, and SNPs genotypes. The patients were classified as positive or negative for a pathogenic variant in the 6 NGS studied genes. The statistical analysis was performed with R-software (version 4.4.2 for Windows). The logistic regression (Linear Generalised Model) was used to compare the APOE allele and genotype frequencies between the groups, and the mean LDL-C values between the SNPs genotypes. A $p < 0.01$ was considered statistically significant.

## 3. Results

We identified a total of 46 pathogenic variants in 139 of the total 431 patients (32%). No patient had the pathogenic variant in *PCSK9*, and one patient was homozygous for *LDLRAP1* c.207delC (p.Ala70ProfsTer19). This patient was a 31-year-old male with an LDL-C of 648 mg/dL, and the two parents were normo-lipaemic and heterozygous carriers for the variant. Most of the pathogenic variants were in the *LDLR* gene (130 patients), with only 2 and 3 patients harbouring PV in *APOB* and *APOE*, respectively (Table 2). In 292 patients (68%), no pathogenic variant was identified and were thus candidates to investigate the association between common SNPs and the risk of FH.

Notably, three *LDLR* variants were found in 60 patients, representing 43% of the patients with a PV and 14% of the total patients: c.1775G>A (p.Gly592Glu) in 12 patients, c.1285 G>A (p.Val429Met) in 23 patients, and c.2389+4A>G (IVS16+4A>G; intron 16 splicing) in 27 patients. Three patients were heterozygous carriers of the *APOE* c.500_502del (p.Leu167del) deletion. These three patients had LDL-C values of 316, 425, and 443 mg/dL, well above the mean value for the PV-positive patients (298 mg/dL). One patient was a heterozygous carrier for *APOE* c.83 C>T (p.Pro28Leu), described as pathogenic in hypercholesterolemic patients by some authors. This patient was a 62-year-old female with total cholesterol = 326 and LDL-C = 187 mg/dL.

**Table 2.** Pathogenic/likely pathogenic variants found in the FH patients.

| Gene/Variant | Npatients |
|---|---|
| APOB: c.10580G>A (p.Arg3527Gln) | 2 |
| APOE: c.500_502del (p.Leu167del) | 3 |
| APOE: c.83C>T(p.Pro28leu) | 1 |
| LDLR: c.91G>T (p.Glu31Ter) | 2 |
| LDLR: c.251C>T (p.Pro84Leu) | 1 |
| LDLR: c.409G>A (p.Gly137Ser) | 1 |
| LDLR: c.464G>A (p.Cys155Tyr) | 2 |
| LDLR: c.487C>T (p.Gln163Ter) | 1 |
| LDLR: c.520G>A (p.Glu174Ter) | 1 |
| LDLR: c.530C>T (p.Ser177Leu) | 3 |
| LDLR: c.676T>C (p.Ser226Pro) | 1 |
| LDLR: c.682G>A (p.Glu228Lys) | 1 |
| LDLR: c.758G>A (p.Arg253Gln) | 1 |
| LDLR: c.682G>A (p.Glu288Lys) | 1 |
| LDLR: c.889A>C (p.Asn297His) | 1 |
| LDLR: c.916_919dupTCAG (p.Asp307Valfs*3) | 1 |
| LDLR: c.1019G>A (p.Cys340Tyr) | 1 |
| LDLR: c.1060G>A (p.Asp354Asn) | 1 |
| LDLR: c.1176C>A (p.Cys392Ter) | 1 |
| LDLR: c.1216C>T (p.Arg406Trp) | 3 |
| LDLR: c.1222G>A (p.Glu408Lys) | 1 |
| LDLR: c.1246C>T (p.Arg416Trp) | 4 |
| LDLR: c.1285G>A (p.Val429Met) | 23 |
| LDLR:c.1609G>T (p.Gly537Ter) | 1 |
| LDLR: c.1690A>C (p.Asn564His) | 1 |
| LDLR: c.1775G>A (p.Gly592Glu) | 11 |
| LDLR: c.1796T>C (p.Leu599Ser) | 4 |
| LDLR: c.1804G>T (p.Glu602Ter) | 1 |
| LDLR: c.1816G>T (p.Ala606Ser) | 2 |
| LDLR: c.1862C>T (p.Thr621Ile) | 1 |
| LDLR: c.1897C>T (p.Arg633Cys) | 2 |
| LDLR:c.1898G>A (p.Arg633His) | 5 |
| LDLR:c.1966C>A (p.His656Asn) | 2 |
| LDLR: c.1973T>C (p.Leu658Pro) | 1 |
| LDLR: c.2096C>T (p.Pro699Leu) | 3 |
| LDLR: c.2099A>G (p.Asp700Gly) | 1 |
| LDLR: c.2177C>T (p.Thr726Ile) | 2 |
| LDLR: c.2397_2405del (p.Val800_Leu802del) | 1 |
| LDLR: c. 7+1G>T (IVS1+1G>T) | 7 |
| LDLR: c.1987+1 G>A (IVS13+1G>A) | 1 |
| LDLR: c.1988−2 A>T (IVS13−2A>T) | 1 |
| LDLR: c.2389+4 A>G (IVS16+4A>G) | 27 |

**Table 2.** *Cont.*

| Gene/Variant | Npatients |
|---|---|
| LDLR: c.313+2 insT (IVS3+2insT) | 5 |
| LDLR: c.1186+5 G>A (IVS8+5G>A) | 1 |
| LDLR: c.1187+1delG (IVS8−1delG) | 1 |
| LDLRAP1: c.207delC (p.Ala70ProfsTer19) HOMOZYGOUS | 1 |

N = number of index patients (non-related) with the variant.

In reference to the *APOE* polymorphisms, the frequencies observed in our controls did not deviate from the Hardy–Weinberg equilibrium, and were close to the reported values among Caucasians (data accessed from www.ensembl.org (accessed on 28 December 2024). One patient was an *APOE*-e2e2 homozygote and showed the typical phenotype for this genotype: hyperlipoproteinemia type III/dysbetalipoproteinemia (OMIM617347) with increased plasma cholesterol and triglycerides, a consequence of impaired apolipoprotein E function that results in reduced clearance of chylomicron and VLDL remnants. Compared to patients with a PV, PV-negative patients showed a significantly higher frequency of *APOE*-e3e4/e4e4 ($p$ = 0.006). Compared to controls, PV-carriers had no significant difference (*APOE*-e3e4/e4e4 17% vs. 13%). A total of 24% and 4% of PV-negatives were *APOE*-e2e4 and e4e4, compared to 12% and 1% of the controls (Table 3). Thus, 29% of the PV-negatives were e3e4/e4e4-carriers, compared to 13% of the controls ($p$ = 0.0002, OR = 2.63, 95% CI = 1.57–4.40). We compared the LDL-C levels between the *APOE* genotypes in the PV-negative patients. The *APOE*-e4e4 patients had significantly higher mean LDL-C compared to the other genotypes (253 vs. 218 mg/dL, $p$ = 0.047) (Figure 1).

**Table 3.** Main values in the FH patients with/without pathogenic variants and population controls. MAF = minor allele frequency.

| | PV POS N = 139 | PV NEG N = 292 | Controls N = 165 |
|---|---|---|---|
| AGE at diagnostic range in years | 18–40 | 25–45 | 18–60 |
| MALE | 76 (55%) | 151 (52%) | 78 (47%) |
| FEMALE | 63 (45%) | 141 (48%) | 87 (53%) |
| LDL-C range mG/dL | 190–750 | 180–358 | 61–150 |
| LDL-C mean mG/dL | 268 | 213 | 132 |
| *APOE*-e2/e3/e4 | | | |
| e2e2 | 0 | 1 | 0 |
| e2e3 | 9 (6%) | 14 (5%) | 26 (16%) |
| e2e4 | 1 | 0 | 2 (1%) |
| e3e3 | 106 (76%) | 193 (66%) | 115 (70%) |
| e3e4 | 23 (17%) | 71 (24%) | 20 (12%) |
| e4e4 | 0 | 13 (4%) | 2 (1%) |
| e3e4 + e4e4 | 23 (17%) | 84 (29%) | 22 (13%) |
| e4 | 0.09 | 0.17 | 0.08 |

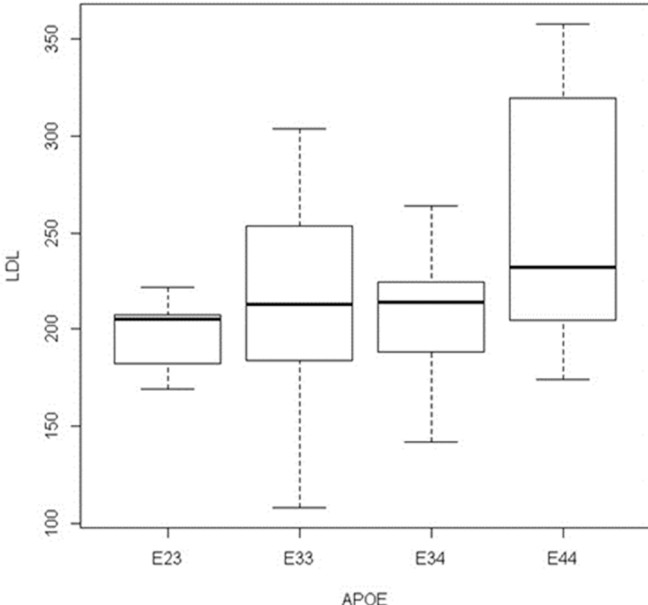

**Figure 1.** Mean LDL-C values according to the *APOE* genotypes in familial hypercholesterolemia patients negative for a pathogenic variant. The *APOE*-e4e4 patients (N = 13) had higher mean LDL-C compared to the other genotypes (253 vs. 218 mg/dL, *p* =0.047).

## 4. Discussion

Approximately half of the patients with a diagnosis of FH did not carry a PV in the "classical" FHgenes. Although the study of other genes related to lipid metabolism could identify additional patients with PVs, most of the cases would remain genetically unclassified. It has been established that many of the FH-negative patients for PVs could carry a combination of common variants nominally associated with a modest increase in LDL-C, and are thus classified as being likely polygenic cases. In recent years, several studies reported risk scores based on the presence of common gene variants associated with LDL-C levels in the general population, including some in Spanish and other European populations [11,13]. However, the clinical translation of these scores is still controversial [19].

The aim of our study was to investigate the association between some of the reported LDL-C risk variants and a diagnostic of FH in the absence of a causative PV. Nominally, only the *APOE*-e3e4 and e4e4 genotypes were significantly increased in the PV-negative patients. In particular, 29% of these patients were e3e4/e4e4 compared to 13% of the population controls. In contrast, e3e3 and e2e3 were less common among the patients. Compared to other studies, the effect of the *APOE*-e3e4/e4e4 genotypes on the risk of FH was much stronger in our cohort. Of note, the frequency of e4e4 was 4% in the patients and 1% in the controls, while the frequency of the e3e4 was double that in the patients (24% vs. 12%). This suggested that ε44 might exhibit the highest risk of hypercholesterolemia among the PV-negative patients.

The role of *APOE* in dyslipaemias has been well established. Some patients with familial dysbetalipoproteinemia type III are homozygous for the e2 isoform, and several *APOE* rare pathogenic variants have been found in patients with familial hypercholesterolemia [20–25]. The p.Leu167del was the only *APOE* pathogenic variant found in our FH patients (three cases). In addition, common *APOE* variants could contribute to polygenic hypercholesterolemia and to the inter-individual variability in blood cholesterol and LDL-C levels in the general population [26–28].

The amino acid change that defines the *APOE*-e4 allele (p.112Arg) is fixed in modern apes and would be the human ancestral allele, with the highest frequency among African Pygmies and other groups from regions where food supply has been historically scarcer

compared to the Mediterranean basin (regions that had a morestable agricultural-based food supply). Some authors have speculated that the *APOE*-e4 allele could have become a risk variant for hypercholesterolemia and its disease-associated conditions under exposure to a Western diet [29,30].

Our study has several limitations. First, we studied the main genes associated with FH, but other genes could be associated with this trait. Also, we did not determine copy-number variations (CNVs, mainly large *LDLR* deletions) that could be found in 5–10% of FH-patients (particularly those with the highest LDL-C levels) [31]. The lack of data on CNVs could reduce the positive detection rate, underestimating the number of patients with PVs [32].

In conclusion, *APOE* pathogenic variants are a rare cause of familial hypercholesterolemia in the Spanish population. However, the *APOE*-e4 allele was significantly over-represented among patients with familial hypercholesterolemia negative for a pathogenic variant in the main FH genes. In particular, the association was highly relevant for *APOE*-e4e4 homozygotes.

**Author Contributions:** Study design: E.C.; patient assessment and data acquisition: all the authors; database: all the authors; genotyping: L.M.V.-P., D.V.-C., C.G.-L., J.G. and E.C.; data filtering and analysis: L.M.V.-P., D.V.-C. and E.C.; statistical analysis: D.V.-C. and E.C.; analysis of results: L.M.V.-P., D.V.-C., R.L. and E.C.; writing the manuscript: E.C.; revision of manuscript: all the authors. All authors have read and agreed to the published version of the manuscript.

**Funding:** This work was supported by a grant from the Spanish Plan Nacional de I+D+I Ministerio de Economía y Competitividad and the European FEDER grants ISCIII-PI21/00971 (E.C.) and RICOR2040- RD21/0005/0011 (E.C.). Daniel VC is the holder of a pre-doctoral grant from ISCIII.

**Institutional Review Board Statement:** Comité de Ética de la investigación con medicamentos de Asturias, code CEimPA 2022-267.

**Informed Consent Statement:** This study was approved by the clinical research ethics committee of Hospital Universitario Central Asturias (HUCA). All the participants gave their consent to participate in this study. The data were handled in observance of Spanish legislation on data protection. This study complies with the principles of the Declaration of Helsinki ("Recommendations guiding doctors in biomedical research involving human subjects").

**Data Availability Statement:** The data that support the findings of this study are available from the corresponding author upon reasonable request. An Excel file with the raw data is available for meta-analysis research.

**Conflicts of Interest:** The authors declare no conflicts of interest.

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
