# Peer review of "Contribution of Rare and Common APOE Variants to Familial Hypercholesterolemia in Spanish Cohort"

_cardiogenetics, doi:10.3390/cardiogenetics15010003_

Round 1
Reviewer 1 Report
Comments and Suggestions for Authors
Dear Authors, I was reviewing with interest the manuscript entitled "Contribution of rare and common APOE variants to familial hypercholesterolemia in a Spanish cohort". Your manuscript deals with a highly interesting subject and you used sufficient methods to investigate a spanish cohort. The APOEe4 allele seems to be an important player in patients with FH. However I have some minor suggestions to improve the manuscript:
You, the authors, don't use stringently the abbreviation "FH". In some cases, there seems to be a wrong variant with "HF".
The first sentence in the Results-section is not clear. From which cohort do these 139 patients come from? Please explain more in detail.
In table 1 there should be also details shown about the main characteristics of the control group and that there are no significant differences between the examined groups.
Additional, you should discuss the interesting results in the context of already published studies about genetic polymorhisms (in FH) in Spain or southern Europe.
Author Response
Dear Authors, I was reviewing with interest the manuscript entitled "Contribution of rare and common APOE variants to familial hypercholesterolemia in a Spanish cohort". Your manuscript deals with a highly interesting subject and you used sufficient methods to investigate a spanish cohort. The APOEe4 allele seems to be an important player in patients with FH. However I have some minor suggestions to improve the manuscript:
You, the authors, don't use stringently the abbreviation "FH". In some cases, there seems to be a wrong variant with "HF": THIS WAS CORRECTED.
The first sentence in the Results-section is not clear. From which cohort do these 139 patients come from? Please explain more in detail. This was revised: 139 of the total 431 patients.
In table 1 there should be also details shown about the main characteristics of the control group and that there are no significant differences between the examined groups. Additional information for the controls is provided in table 1.
Additional, you should discuss the interesting results in the context of already published studies about genetic polymorhisms (in FH) in Spain or southern Europe: There are two studies that addressed the issue of polygenic FH in Southern Europe, among others the Lamiquiz Romero et al and Futema et al. These studies were discussed in the context of polygenic studies, that included the APOE polymorphism as the main genetic determinant in risk-scores.
Reviewer 2 Report
Comments and Suggestions for Authors
The study addresses a significant gap in understanding familial hypercholesterolemia (FH), particularly the contribution of *APOE* variants to FH in patients without identified pathogenic variants (PVs).
Next-generation sequencing (NGS) to analyze vital FH-related genes ensures thorough genetic characterization. Including appropriate controls from the same population enhances the reliability of comparisons.
Identifying *APOE*-e4e4 as a significant risk factor in PV-negative patients is valuable for understanding FH's polygenic nature.
The study adheres to ethical standards with participant consent.
Raw data availability promotes reproducibility and further meta-analysis.
The study did not include copy number variation (CNV) analyses, which could have identified additional PVs. CNVs account for up to 10% of FH cases and could have improved the detection rate.
The exclusion of additional genes potentially linked to lipid metabolism limits the comprehensive understanding of genetic contributions.
The study population is restricted to a Spanish cohort, which may limit the generalizability of findings to other ethnic groups.
The relatively small sample size (292 PV-negative patients and 165 controls) might underrepresent the genetic diversity of FH.
The significance of the observed APOE effects may be overestimated due to the need to adjust for population stratification or other confounders.
The logistic regression analysis does not mention correcting for multiple testing.
Limited exploration of environmental and lifestyle factors, such as diet or exercise, significantly influences lipid profiles.
The study did not stratify patients by treatment history, which could affect LDL-C levels and their association with genetic variants.
While limitations are acknowledged, their potential impact on the study's conclusions must be thoroughly discussed.
The hypothesized evolutionary advantage of the *APOE*-e4 allele lacks direct evidence from the current cohort and might appear speculative without robust backing.
Incorporate CNV testing and broader panels of lipid metabolism-related genes.
xplore epigenetic factors or gene-environment interactions.
Expand the cohort to include diverse populations to enhance generalizability.
Apply corrections for multiple testing and stratify analyses by potential confounders such as age, sex, and treatment.
Incorporate PRS better to elucidate the polygenic contributions to LDL-C levels and FH.
Collect detailed lifestyle and treatment data to improve phenotypic correlations.
This study contributes valuable insights into the role of *APOE* variants in FH, particularly for PV-negative cases. However, methodological and population-related limitations restrict its broader applicability. Future research should aim for a more holistic approach by integrating genetic, environmental, and clinical factors to provide a more comprehensive understanding of FH.
Author Response
"The study addresses a significant gap in understanding familial
hypercholesterolemia (FH), particularly the contribution of *APOE*
variants to FH in patients without identified pathogenic variants (PVs).
Next-generation sequencing (NGS) to analyze vital FH-related genes
ensures thorough genetic characterization. Including appropriate
controls from the same population enhances the reliability of comparisons.
Identifying *APOE*-e4e4 as a significant risk factor in PV-negative
patients is valuable for understanding FH's polygenic nature.
The study adheres to ethical standards with participant consent.
Raw data availability promotes reproducibility and further meta-analysis.
The study did not include copy number variation (CNV) analyses, which
could have identified additional PVs. CNVs account for up to 10% of FH
cases and could have improved the detection rate.
Resp: this was indicated in the ms as a limitation. We have data about the CNVs detected with MLPA in a very limited number of patients, all negative. Because the whole cohort was not studied by this technique the data are not presented.
The exclusion of additional genes potentially linked to lipid metabolism
limits the comprehensive understanding of genetic contributions.
Resp: we agree and indicate this in the discussion as a limitation. However, the APOE polymorphisms is the main polygenic determinant and any other variants should be analysed in the APOE-genotype context to build a risk-score
The study population is restricted to a Spanish cohort, which may limit
the generalizability of findings to other ethnic groups. The relatively small sample size (292 PV-negative patients and 165 controls) might underrepresent the genetic diversity of FH. The significance of the observed APOE effects may be overestimated due
to the need to adjust for population stratification or other confounders. The logistic regression analysis does not mention correcting for multiple testing.
Resp: the statistical analysis compared the frequency of the allele genotype frequencies between patients with/without a PV. In the LGM we included sex and age as covariates. Considering multiple testing, the p value could be reduced to 0.01 (instead of p<0.05). this was indicated in the methods.
Limited exploration of environmental and lifestyle factors, such as diet
or exercise, significantly influences lipid profiles. The study did not stratify patients by treatment history, which could affect LDL-C levels and their association with genetic variants. While limitations are acknowledged, their potential impact on the
study's conclusions must be thoroughly discussed.
Resp: The lipid values corresponded to pretreatment with lipid lowering drugs, according to the clinical history. This was indicated in the methods.
The hypothesized evolutionary advantage of the *APOE*-e4 allele lacks
direct evidence from the current cohort and might appear speculative
without robust backing.
Resp: we revised this in the discussion, indicating that there is a lack of robust evidences for the hypothesis, other than the association with lipids and the different worldwide distribution of APOE-e4.